# The Effect of the Inclusion of Different Concentrates in Feed Rations on the Contents of Tocopherols, β-Carotene and Retinol in the Livers and Longissimus dorsi Muscles of Farm-Raised Fallow Deer (*Dama dama* L.)

**DOI:** 10.3390/ani12233311

**Published:** 2022-11-27

**Authors:** Sylwia Czurgiel, Zofia Antoszkiewicz, Magdalena Mazur-Kuśnirek, Marek Bogdaszewski

**Affiliations:** 1Department of Animal Nutrition and Feed Science, University of Warmia and Mazury in Olsztyn, 10-718 Olsztyn, Poland; 2Institute of Parasitology of the Polish Academy of Sciences, Research Station in Kosewo Górne, 11-700 Mrągowo, Poland

**Keywords:** fallow deer, tocopherols, retinol, β-carotene, liver, longissimus dorsi muscle

## Abstract

**Simple Summary:**

In many countries, the main source of venison is game shooting, but deer farming is growing intensively, partly as a consequence of greater interest in alternative meat species. The meat of cervids, including fallow deer, has a desirable, beneficial basic chemical composition, but there is little information on the concentrations of biologically active compounds (β-carotene, retinol and tocopherols). In the present study, we evaluated the contents of β-carotene, retinol and tocopherols, as well as the basic chemical compositions, of the meat and livers of fallow deer fed rations based on hay with different concentrated feeds (oat grain, protein concentrate and pellets). More retinol was contained in the livers and meat of fallow deer fed rations with protein concentrate or pellets (experimental groups) than in the livers and meat of fallow deer fed rations with oat grain (the control group). The proportions of the different concentrated feeds did not affect the basic chemical compositions and tocopherol contents in the fallow deer meat and livers. Based on our results, it was concluded that protein concentrate and pellets can be valuable components in the rations of farmed fallow deer.

**Abstract:**

This study analyzed the chemical composition and contents of β-carotene, tocopherols and retinol in the longissimus dorsi (LD) muscles and livers of fallow deer (*Dama dama* L.) fed diets based on hay containing different concentrates (hay and oats—HO; hay, oats and protein concentrate—HOP; hay and pellets—HP). The provitamin and lipophilic vitamin contents in the samples of feed and animal tissues were determined by high-performance liquid chromatography (HPLC). The contents of retinol were nearly two-fold higher (*p* < 0.002) in the livers and many-fold higher in the LD muscles of fallow deer fed the HOP and HP diets. β-carotene was not identified in the livers or LD muscles of fallow deer. The concentrations of α-tocopherol and total tocopherols were higher in the livers and LD muscles of the animals that received the HO and HP diets, but the noted differences were not significant. The inclusion of various concentrates in the feed rations had no influence on the proximate chemical compositions of the livers or LD muscles of fallow deer.

## 1. Introduction

In Poland, most game meat is harvested through hunting [1]. The game meat market undergoes seasonal changes and is affected by the hunting intensity for different species and age and sex groups [1]. The number of cervid farms has increased in recent years [2]. Deer farming is popular in New Zealand (with an estimated population of 2.5 million head), Australia, China, Canada, the USA and Scandinavian countries [3]. Cervid farms have also begun to develop in Poland and other European countries, such as Czechia, France and Belgium [4,5,6], and the fallow deer (*Dama dama* L.) is one of the most commonly raised species [7]. According to meat consumers, game meat is organic, safe and natural [8]. Annual game meat consumption varies across Europe [9], and it remains low in Poland (80 g per capita, compared with 600 g per capita in Germany) [1]. Due to its highly desirable chemical composition, game meat can be an alternative protein source in the human diet [1]. The proximate chemical composition of fallow deer meat has been relatively well researched, but little is known still about the contents of biologically active compounds (including β-carotene, retinol and tocopherols); only scarce information is also available on the nutritional value of offal from game animals (including fallow deer), and most published studies have focused on the risk of contamination [7,10,11,12]. Vitamin A and its precursor, β-carotene, as well as tocopherols (vitamin E), are essential nutrients that play very important roles in numerous biological processes. They are fat-soluble and share several common mechanisms that support their metabolism and transfer to offspring. Both vitamin A and vitamin E are potent antioxidants and immune system boosters; they support tissue protection, reproduction, growth and development [13]. These compounds cannot be synthesized in the animal body and they have to be supplied by feed. Information on the nutritional management of cervids is limited; therefore, the optimal combination of nutrient sources in their diets has not yet been established [14]. In Poland, farmed cervids graze pasture in spring and summer and receive preserved fodder (hay, haylage, silage, ground grain) in fall and winter [15]. Concentrate feeding is limited to periods of increased nutrient demand (winter and antler growth and lactation periods) [15].

The aim of this study was to evaluate the effects of different concentrates (protein concentrate and pellets) in diets based on hay on the proximate chemical composition and contents of β-carotene, tocopherols and retinol in the longissimus dorsi (LD) muscles and livers of farm-raised fallow deer (*Dama dama* L.).

## 2. Materials and Methods

The fallow deer used in this experiment were managed in accordance with the Act of 15 January 2015 on the protection of animals used for scientific or educational purposes [16].

### 2.1. Animals, Experimental Design and Diets

A total of 36 (n = 36) fallow deer (*Dama dama* L.) at around 10 months of age were divided into three equal groups (3 × 12). The animals were raised on a farm owned by the Institute of Parasitology of the Polish Academy of Sciences, Research Station in Kosewo Górne (Poland, 53°N 21°E). They were housed indoors in groups, with no access to yards; feed and water were available ad libitum. The experiment lasted 60 days, including a 21-day adaptation period. In each group, the feed rations consisted of hay (400 g/head) and a mineral–vitamin premix (20 g/head) (LNB—Cargill Ltd., Kiszkowo, Poland). In the control group (HO), the fallow deer were fed hay and ground oat grain (400 g/head); in the HOP group, the animals were fed hay, ground oat grain (300 g/head) and protein concentrate (56 g of soybean meal + 56 g of rapeseed meal; 112 g/head) (EKO-PASZ Feed Mill Zbigniew Kropiewnicki, Mońki, Poland); and in the HP group, they were fed hay and pellets (500 g/head) (Piast Pasze Ltd., Lewkowiec, Poland). The roughage-to-concentrate ratio in the feed rations was 40:60 on a dry matter (DM) basis (Table 1). The chemical compositions of the feeds and diets are presented in Table 2. The feed rations were thoroughly mixed, and representative samples were taken to determine their chemical compositions. The mixed rations were fed twice a day: in the morning, at 7 a.m., at 70% of the ration weight, and in the afternoon, at 3 p.m. (the remaining 30% of the ration weight). The weight of the rations was planned to leave 5–10% uneaten by each animal. After 60 days of the feeding trial, the animals were slaughtered and samples of liver tissue and LD muscle were collected. Slaughter was carried out on the farm, under veterinary supervision, in accordance with the Regulation of the Ministry of Agriculture and Rural Development of 9 September 2004 on the qualifications of persons competent to carry out slaughter and the requirements and methods for slaughtering and killing animals [17]. The animals were held still in a crush, stunned with a captive bolt gun (Radical) and bled by incising a carotid artery. The carcasses were transported to a meat processing plant (in a refrigerator truck, at −4 °C; time of transport: approx. 1 h), where they were subjected to veterinary examination following evisceration and skin removal. The chilled carcasses (2–3 °C, 48 h) were dressed.

### 2.2. Sampling and Analysis

Samples of the LD muscle were cut out of the right side of each carcass (at the last rib; 36 samples in total). Muscle and liver tissue samples were packaged in polyethylene bags, transported to the laboratory in isothermal containers on ice and frozen until analysis (for approximately one week). Samples of feeds, diets, liver tissues and LD muscles (thawed prior to analysis) were dried at a temperature of 60 °C (Binder GmbH laboratory oven, Tuttlingen, Germany) and ground to a particle size of around 1 mm (ZM 200 ultra centrifugal mill, Retsch, Haan, Germany), and their proximate chemical compositions were determined by standard AOAC methods [18]. The neutral detergent fiber (NDF) contents of the feed and diet samples were determined by the method proposed by Van Soest et al. [19] using the ANKOM220 fiber analyzer (ANKOM Technology Corp., Macedon, NY, USA). The contents of β-carotene, retinol and tocopherols were determined under limited exposure to sunlight for fresh (non-dried) feed and diet samples that had been ground with a blender (Robert Bosch GmbH, Gerlingen, Germany) and for liver tissue and LD muscle samples that had been thawed prior to analysis. Feed and diet samples (all weighing 5 g) were extracted with a mixture of petroleum ether and acetone (1:1 *v*/*v*) at room temperature, in the dark, for 18 h, and next they were saponified with ethanolic 10% KOH solution (2 h in the dark, under nitrogen atmosphere; vortex—IKA Poland Ltd., Warsaw, Poland). Saponified samples were extracted with ethanol (96%) and then repeatedly with petroleum ether. The extracts were washed with 10% aqueous NaCl solution and repeatedly with deionized water; the eluates were dehydrated with anhydrous sodium sulfate and evaporated to dryness at a temperature of 40 °C in a rotary evaporator (IKA Poland Ltd., Warsaw, Poland). Homogenates of fresh liver tissue and LD muscle samples (1 g) (Ultra-Turrax T25 homogenizer, Janke&Kundel IKA-Labortechnik, Staufen, Germany) were saponified with 2 cm^3^ of a saponification mixture (375 g KOH, 750 cm3 H2O, 450 cm^3^ methanol) in a shaking water bath (200 rpm, 80 °C, 15 min, Julabo GmbH, Seelbach, Germany). The samples were cooled, 5 cm^3^ of 35% ethanol and 5 cm^3^ of 10% NaCl were added and the samples were shaken (vortex—IKA Poland Ltd., Warsaw, Poland) and left for 5 min, after which 4 cm^3^ of n-hexane was added and the samples were shaken again and centrifuged at 1500× *g* for 15 min (MPW Med. Instruments, Warsaw, Poland). A quantity of 3 cm^3^ of the extract was collected from the supernatant, and extraction with 4 cm^3^ of n-hexane was repeated. Combined extracts were evaporated to dryness under a stream of N2. The samples were dissolved in 1 cm^3^ of n-hexane and filtered through a syringe filter [20,21,22]. The contents of tocopherols and retinol in the hexane-extracted samples of feeds, diets, liver tissue and LD muscle samples were determined by reversed-phase high-performance liquid chromatography (RP-HPLC) (Shimadzu, Kyoto, Japan): column—Nucleosil C18; mobile phase—methanol:H2O (95:5 *v*/*v*); flow rate—1 cm^3^ min^−1^; tocopherol detection—RF detector Ex 293 and Em326; external standards: (±)-a-tocopherol (DL-all-rac a-tocopherol), b-tocopherol, (+)-g-tocopherol, (+)-d-tocopherol (Sigma-Aldrich, St. Louis, USA) [23]; retinol detection—UV-VIS 336 nm; external standard—Retinol, synthetic (SIGMA-ALDRICH) [24]. The contents of β-carotene in the samples was determined by HPLC (Shimadzu, Kyoto, Japan): column—Phenomenex Gemini 5μm, C18, 110 Å, 250 × 4 mm; mobile phase—acetonitrile:methanol:dichloromethane (750:200:50 *v*/*v*/*v*) (HPLC grade SIGMA-ALDRICH); flow rate—1 cm^3^ min^−1^; UV-vis 450 nm detector, 20 μL loop; external standard—β-Carotene Type I, synthetic [25].

### 2.3. Statistical Analysis

The results were processed statistically by one-way ANOVA, linear correlation analysis and Duncan’s test. Arithmetic means and standard errors of the means (SEMs) were calculated. Results were regarded as significant at *p* ≤ 0.01 and *p* ≤ 0.05. All calculations were performed using STATISTICA version 13.3 software.

## 3. Results

The proximate chemical compositions (DM, CA, CP and EE) of the liver tissue samples collected from the fallow deer were similar for all groups (Table 3). The retinol contents of the liver tissue samples were nearly two-fold higher in the HP group (hay + pellets) (by approx. 47.5 mg/kg) and the HOP group (hay + oat grain + protein concentrate) (by approx. 36.8 mg/kg) compared to the control group (HO; hay + oat grain, *p* < 0.01) (Table 3). A statistical analysis did not reveal differences in the contents of α-, β-, γ- or δ-tocopherol or total tocopherols in the liver tissue samples. The liver tissue samples collected from the control group (HO) animals had higher concentrations of α-tocopherol and total tocopherols than those taken from animals fed the HOP diet and higher β-tocopherol contents than those taken from the HOP and HP groups, but the observed differences were not significant. 

The contents of DM, CA and CP in the LD muscle samples were similar for all groups. The concentrations of EE in LD muscle samples were three- to four-fold higher in the control group (HO) compared to the HOP and HP groups (*p* < 0.01; Table 4). The tocopherol contents in the LD muscle samples were comparable for all groups. However, the concentrations of α-tocopherol and total tocopherols in the LD muscle samples were somewhat higher (non-significant differences) in the fallow deer fed the HP diet than in the deer that received the HO and HOP diets (non-significant differences). Samples of the LD muscles did not contain β-carotene. Retinol concentrations in the LD muscle samples were many-fold higher in group HP compared to the other groups, while the LD muscles of the control group animals had the lowest retinol contents. 

The analysis showed some significant correlations between the vitamin contents in the livers and those in the diets (Table 5). The strongest correlations obtained were between α-tocopherol and γ-tocopherol (very strongly negative), α-tocopherol and total tocopherols (very strongly positive), as well as γ-tocopherol and total tocopherols (very strongly negative). Noteworthy is the strong negative correlation found between β-tocopherol contents in the livers and the concentrations of retinol in the feed rations (r = −0.81; *p* < 0.05). At the same time, the analysis showed a moderate positive correlation of liver α-tocopherol with dietary retinol. A weak, not statistically significant correlation occurred between β-tocopherol and γ-tocopherol, as well as total tocopherols and retinol.

α-tocopherol contents in the longissimus dorsi muscle samples from the fallow deer were strongly positively correlated with total tocopherol contents in the rations (Table 6). Moderate but positive and significant correlations were found between β-tocopherol contents in meat and dietary retinol and between β-tocopherol and total tocopherols (both r = 0.63). Slightly lower, moderate significant coefficients were determined for total tocopherols and retinol and for α-tocopherol and β-tocopherol.

## 4. Discussion

### 4.1. Chemical Composition, Contents of β-Carotene, Retinol and Tocopherols in the Livers of Fallow Deer, and Correlations with Selected Biologically Active Compounds

Livestock offal contains essential nutrients, such as vitamins, proteins, minerals and fat, which are comparable to those contained in meat [26,27]. The quality of meat products is influenced by both genetic factors and, to a large extent, environmental factors, including nutrition [27]. In the present study, diets of different compositions (hay + oat grain, hay + oat grain + protein concentrate and hay + pellets) with comparable nutrient contents fed to fallow deer did not induce differences in the contents of DM, CA, CP or EE in the deer livers. In these study feed intake was not recorded as in the approach used by some other authors [3]; therefore, our study was useful in assessing the effects on β-carotene, retinol and tocopherol contents in the livers and LD muscles of fallow deer. Animal liver is a good source of protein, as confirmed by the results of this study (22–23%), with quite a low fat content (1.3–1.6%). The chemical compositions of fallow deer livers were similar to those of other animal species, including sheep, calves and cattle [27,28]. The results of the studies by Borowiec et al. [28] and Biel et al. [27] indicated higher fat contents (about 3.6%) in samples of sheep livers fed hay and concentrate and in liver samples from calves, beef cattle and lambs that were fed and maintained under organic-production conditions.

Similar to retinol, vitamin E is stored in the liver [29]. In the evaluated liver tissue samples collected from fallow deer, α-tocopherol was the predominant form of vitamin E (0.34–0.65 mg/kg). According to Brigelius-Flohe [30], α-tocopherol accumulates in the liver due to the specific selection of RRR-α-tocopherol by the hepatic α-tocopherol transfer protein (αTTP). The remaining tocopherol fractions (β-, γ- and δ-) provided by feed are excreted in feces or secreted into bile. The available information on the contents of tocopherols and retinol in the livers of cervids, including fallow deer, is limited. The concentrations of α-tocopherol in the livers of ruminants reported in the literature are many-fold higher than those noted in the present study. The concentration was estimated at 10 mg/kg FW by Puls [31]. In a study by Humann-Ziehank et al. [32], it reached 12–17 mg/kg in the livers of sheep and 30–41 mg/kg in the livers of roe deer, while Rodríguez-Estival et al. [29] noted 14 mg/kg of α-tocopherol in the livers of wild ungulates. Yang et al. [33] demonstrated that a higher vitamin E concentration in the liver was correlated with higher dietary intake. In this study, correlation analysis showed a very strong positive correlation between total tocopherols in the dietary rations and liver α-tocopherol concentrations. The concentrations of α-tocopherol and total tocopherols in the liver were higher in fallow deer fed the HP diet (containing pellets), which was the richest in tocopherols, than in those fed the control HO diet (by approx. 37% and 48%, respectively) and the HOP diet (containing protein concentrate; by approx. 19% and 39%, respectively). The noted values were not significant, probably due to insufficient concentrations of tocopherols in the feed rations. It should be stressed, however, that a higher content of vitamin E (all forms and total tocopherols) in the HOP diet as compared with the HO diet did not contribute to increasing vitamin E concentrations in the livers of HOP group animals, perhaps because there was too small a difference between tocopherol levels in these doses. 

In the present study, β-carotene was not identified in liver tissue samples (Table 3). β-carotene is rapidly converted into retinol in the livers of many animal species, including cervids [34]. Due to considerable differences in carotenoid metabolism between animal species, Schweigert [35] divided them into “white-fat” animals that absorb carotenoids at very low levels or do not absorb them at all (goats, sheep, pigs, rodents) and “yellow-fat” animals that can absorb carotenoids (cattle, horses, birds). Based on the results of the present study (absence of β-carotene), fallow deer could be classified in the first group. 

As noted by Majchrzak et al. [36], vitamin A concentrations in the livers of livestock species may vary widely. Vitamin A contents ranged from 1.1 to 6.7 mg retinol equivalents/100 g in cattle [36] and were found to be 136 mg/kg in pasture-fed goats and 484 mg/kg in pasture-fed sheep [37]. In the present study, the fallow deer livers contained 49.12–96.89 mg/kg of retinol. The highest concentrations of retinol were contained in the livers of fallow deer fed HP rations, as a result of these rations having the highest β-carotene contents. Liver tissue samples collected from the fallow deer fed the HOP diet contained considerably higher levels of retinol (85.89 mg/kg in the HOP group vs. 49.12 mg/kg in the HO group), despite comparable concentrations of provitamin A (β-carotene) in the respective diets (22.54 mg kg^−1^ DM in the HO diet and 23.50 mg kg^−1^ DM in the HOP diet). In this regard, research should continue.

### 4.2. Chemical Composition and Contents of β-Carotene, Retinol and Tocopherols in the LD Muscles of Fallow Deer and Correlations with Selected Biologically Active Compounds

The proximate chemical composition of red deer and fallow deer meat has been extensively researched [7]. Numerous studies have investigated the chemical composition of meat from wild and farmed cervids [5,10], both male and female [38], which is affected by such factors as muscle type [39] and nutrition [40]. It is noteworthy that, irrespective of differences in experimental conditions, game meat is characterized by a high protein content and a low concentration of intramuscular fat [40]. The results of the present study correspond with this finding. Fallow deer LD muscle samples contained around 200–210 g/kg of protein and 2–9 g/kg of fat (*p* = 0.002). Daszkiewicz et al. [5] reported that meat from wild and farmed fallow deer contained around 22.5% of protein and 0.2–0.5% of fat and stressed that in farm-raised deer the fat content of meat increases with age and feeding intensity. 

In the current study, the LD muscle samples collected from fallow deer contained 0.94–1.15 mg/kg of total tocopherols. There is a scarcity of published data on the tocopherol contents of fallow deer meat. Tocopherol concentrations in meat from other cervid species reported in the literature are higher than those determined in this experiment. For instance, the contents of vitamin E in the LD muscles were 4.4–5.6 mg/kg in pasture-finished red deer [41] and 5.36–5.5 µg/g in reindeer fed pelleted feed mixtures as complete diets [42]. Numerous studies have shown that in ruminants the vitamin E content of meat is directly correlated with dietary vitamin E intake [43,44,45]. Correlation analysis in this study showed a positive, very strong relationship between the concentration of α-tocopherol in meat and the content of total tocopherols in the rations. The concentrations of α-tocopherol and total tocopherols in the LD muscle were higher in fallow deer fed the HP diet, which was richer in α-tocopherol than the HO and HOP diets (Table 2), by around 5% and 20% compared with the HO group and by around 3.5% and 18% compared with the HOP group (Table 4).

It is well known that there are differences in the metabolism and accumulation of both carotenoids and retinoids in mammals [35,37]. However, the deposition of these compounds in tissues depends primarily on diet [46] and other factors, such as supplementation levels [47] and the interactions between carotenoids and tocopherols [33]. In addition, Alosilla et al. [48] reported that vitamin A availability is limited in ruminants due to losses by ruminal destruction, which is especially high when ruminants are fed high-concentrate diets. Published data on retinol concentration in cervid meat are scarce and contradictory. Dannenberger et al. [49] demonstrated that the concentrations of vitamin A in the muscles of wild roe deer were below the detection limit of 0.01 mg/kg. Retinol concentrations in muscles were 0.026–0.030 µg/g in pasture-fed red deer and 0.029–0.040 µg/g in red deer that received pelleted diets [42]. An experiment performed on suckling lambs revealed that retinol concentrations in meat were higher in lambs fed milk replacer than in lambs fed maternal milk (12.8 vs. 5.8 µg/100 g), which resulted from the higher vitamin A content in the milk replacer [50]. In the current study, retinol concentrations were moderately but positively correlated with dietary α-tocopherol and total tocopherols. The retinol content of the LD muscle was many-fold lower in fallow deer fed the HO and HOP diets (0.08 and 0.31 mg/kg, respectively) than in those receiving the HP diet (2.43 mg/kg). It should be noted that the HP diet was approximately twice as rich in β-carotene (provitamin A) as the other diets (43.39 vs. 22.54 and 23.50 mg/kg DM). 

The accumulation of carotenoids varies widely across animal species [46]. β-carotene is the predominant carotenoid in the adipose tissue and blood serum of cattle, whereas lutein predominates in sheep and goats [37,51,52]. In sheep, β-carotene was detected in trace amounts only, since it is almost entirely converted to retinol following absorption [46]. Data on carotenoid concentrations in the tissues of cervids are limited, and research focuses on the antioxidant activity of β-carotene and its effect on meat quality, mostly in cattle [46]. In the current study, the concentrations of β-carotene in the LD muscles of fallow deer were below the detection limit, which suggests that it is mostly converted into retinol. In contrast, the β-carotene content of meat was 0.18 µg/g in cows fed a maize-based diet and 0.19–0.33 µg/g in β-carotene-supplemented animals [53]. Similar concentrations of β-carotene (0.116–0.152 µg/g) were determined by Walshe et al. [54] in the LD muscles of steers reared under organic and conventional production systems.

## 5. Conclusions

β-carotene was not identified in the livers or LD muscles of fallow deer. Retinol contents were higher in liver tissue samples from fallow deer fed diets with higher contents of β-carotene supplied by concentrates. The inclusion of concentrates in the feed rations did not induce significant differences in the contents of tocopherols in the LD muscles or livers. Diets richer in tocopherols contributed to higher concentrations of total tocopherols in the LD muscles and livers, and a very strong correlation was found between them. Based on our results, protein concentrate and pellets can be valuable feed components for farmed fallow deer.

## Figures and Tables

**Table 1 animals-12-03311-t001:** Ingredient compositions of diets fed to fallow deer (% DM).

Specification	Group		
	HO	HOP	HP
Ingredients, % DM			
Hay	40	40	40
Oat grain	60	40	-
Protein concentrate	-	20	-
Pellets	-	-	60
Mineral–vitamin premix (g/d, head)	20	20	20

Groups: HO—hay + oat grain; HOP—hay + oat grain + protein concentrate; HP—hay + pellets. Mineral–vitamin premix (g/kg): Ca—21%, P—36%, Na—10%, Mg—4%, Mn—3000 mg, Zn—6000 mg, Fe—4000 mg, Cu—1000 mg, Co—25 mg, I—100 mg, Se—25 mg, vitamin A—500,000 IU, vitamin D—100,000 IU, vitamin E—1000 IU, equivalent of vitamin E—275 IU, vitamin B1—4000 IU. Pellets: dehydrated grass, barley, compressed and molassed beet pulp, wheat gluten, sunflower meal, oats, soybean meal, rapeseed meal, dehydrated alfalfa, vegetable oil, sodium chloride.

**Table 2 animals-12-03311-t002:** Proximate chemical compositions (g kg^−1^ DM) and contents of tocopherols, vitamin A and β-carotene (mg kg^−1^ DM) in feeds and diets fed to fallow deer.

Item	Feed				Group		
	Hay	Oat	Concentrate	Pellets	HO	HOP	HP
Dry matter-DM (g kg^−1^)	866.4	905.1	898.7	906.7	888.5	887.8	891.6
CA	46.2	24.9	44.1	80.1	33.4	37.2	66.5
CP	91.0	122.1	190.4	170.3	109.6	123,3	138,6
EE	15.9	36.2	31.9	33.4	28.0	27.2	26.4
CF	347.3	106.6	114.3	147.9	202.9	204.4	227.6
NDF	286.5	356.8	281.3	347.0	328.7	313.6	322.8
α-tocopherol	1.53	2.81	3.26	5.16	1.93	3.28	4.08
β-tocopherol	1.00	1.11	2.22	8.10	1.40	1.92	6.06
γ-tocopherol	0.66	1.83	4.80	1.16	1.96	3.29	3.23
δ-tocopherol	0.59	0.73	0.80	0.69	1.39	2.00	3.65
Total tocopherols	3.78	6.48	11.08	15.11	6.68	10.49	17.02
Retinol	-	-	150.0	150.0	3.00	3.00	3.00
β-carotene	54.24	0.60	2.57	23.88	22.54	23.50	43.39

Groups: HO—hay + oat grain; HOP—hay + oat grain + protein concentrate; HP—hay + pellets. DM—dry matter, CA—crude ash, CP—crude protein, EE—ether extract, CF—crude fiber, NDF—neutral detergent fiber.

**Table 3 animals-12-03311-t003:** Proximate chemical compositions (g/kg) and contents of tocopherols, β-carotene and retinol in liver tissue samples collected from fallow deer fed different diets (mg/kg).

Item	Group			SEM	*p*-Value
	HO	HOP	HP		
DM	291.1	302.0	290.5	0.311	0.264
CA	14.03	13.97	13.83	0.016	0.898
CP	218.1	229.6	222.0	0.227	0.092
EE	15.33	16.10	13.33	0.060	0.148
α-tocopherol	0.41	0.34	0.65	0.061	0.081
β-tocopherol	0.15	0.05	0.09	0.020	0.123
γ-tocopherol	0.04	0.06	Nd	0.012	0.098
δ-tocopherol	Nd	Nd	Nd	-	-
Total tocopherols	0.60	0.45	0.74	0.064	0.189
β-carotene	Nd	Nd	Nd	-	-
Retinol	49.12 ^A^	85.89 ^B^	96.59 ^B^	6.584	0.002

Nd—not detected. Different superscripts in the same row are significant or trending (A/B: *p* ≤ 0.01). SEM—standard error of the mean. Groups: HO—hay + oat grain; HOP—hay + oat grain + protein concentrate; HP—hay + pellets. DM—dry matter, CA—crude ash, CP—crude protein, EE—ether extract, NDF—neutral detergent fiber.

**Table 4 animals-12-03311-t004:** Proximate chemical compositions (g/kg) and contents of tocopherols, β-carotene and retinol in samples of the longissimus dorsi muscle collected from fallow deer fed different diets (mg/kg).

Item	Group			SEM	*p*-Value
	HO	HOP	HP		
DM	234.5	226.9	225.5	0.227	0.239
CA	11.03	10.87	10.70	0.008	0.278
CP	214.30	210.27	199.53	0.404	0.347
EE	8.60 ^B^	3.03 ^A^	2.30 ^A^	0.106	0.002
α-tocopherol	0.98	0.83	1.03	0.073	0.253
β-tocopherol	0.13	0.10	0.11	0.009	0.369
γ-tocopherol	Nd	0.01	0.01	0.005	0.215
δ-tocopherol	Nd	Nd	Nd	-	-
Total tocopherols	1.11	0.94	1.15	0.081	0.289
β-carotene	Nd	Nd	Nd	-	-
Retinol	0.08	0.31	2.43	0.466	0.066

Nd—not detected. Different superscripts in the same row are significant or trending (A/B: *p* ≤ 0.01). SEM—standard error of the mean. Groups: HO—hay + oat grain; HOP—hay + oat grain + protein concentrate; HP—hay + pellets. DM—dry matter, CA—crude ash, CP—crude protein, EE—ether extract, NDF—neutral detergent fiber.

**Table 5 animals-12-03311-t005:** Coefficients of correlations between feed contents of tocopherols, β-carotene and retinol and contents of tocopherols, β-carotene and retinol in the liver samples collected from the fallow deer.

Item	α-Tocopherol	β-Tocopherol	γ-Tocopherol	δ-Tocopherol	Total Tocopherols	β-Carotene	Retinol
α-tocopherol	1.00						
β-tocopherol	0.10	1.00					
γ-tocopherol	−0.99 *	−0.22	1.00				
δ-tocopherol	-	-	-	1.00			
Total tocopherols	0.95 *	0.42	−0.98 *	-	1.00		
β-carotene	-	-	-	-	-	1.00	
Retinol	0.50 *	−0.81 *	−0.40	-	0.20	-	1.00

** p* < 0.05; r < 0.2—no correlation, 0.2–0.4—weak correlation, 0.4–0.7—moderate correlation, 0.7–0.9—fairly strong correlation, r > 0.9—very strong correlation.

**Table 6 animals-12-03311-t006:** Coefficients of correlation between feed contents of tocopherols, β-carotene and retinol and contents of tocopherols, β-carotene and retinol in samples of the musculus longissimus dorsi collected from the fallow deer.

Item	α-Tocopherol	β-Tocopherol	γ-Tocopherol	δ-Tocopherol	Total Tocopherols	β-Carotene	Retinol
α-tocopherol	1.00						
β-tocopherol	0.58 *	1.00					
γ-tocopherol	−0.28	−0.95	1.00				
δ-tocopherol	-	-	-	1.00			
Total tocopherols	0.99 *	0.63 *	−0.34	-	1.00		
β-carotene	-	-	-	-	-	1.00	
Retinol	0.63 *	−0.28 *	0.57	-	0.58 *	-	1.00

* *p* < 0.05; r < 0.2—no correlation, 0.2–0.4—a weak correlation, 0.4–0.7—moderate correlation, 0.7–0.9—fairly strong correlation, r > 0.9—very strong correlation.

## Data Availability

The data presented in this study are available from the corresponding author.

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
