# Peer review of "The Effect of the Inclusion of Different Concentrates in Feed Rations on the Contents of Tocopherols, β-Carotene and Retinol in the Livers and Longissimus dorsi Muscles of Farm-Raised Fallow Deer (Dama dama L.)"

_animals, 2022, doi:10.3390/ani12233311_

Round 1

Reviewer 1 Report (Previous Reviewer 2)

The new version of this manuscript was significantly improved. We suggested some minor revisions.

Lines 233 and 289. "Correlations between" replace by "correlations with".

Lines 239-240. Could you add a sentence mentioning that feed intake was not evaluated, but your approach was useful in assessing the effect of the diets?

For example. "feed intake was not recorded, the same approach was used by some other authors (Kudrnáčová et al. 2019); therefore, our study was useful to assess the effect of β-carotene, retinol and tocopherols in the liver and  the LD muscle of fallow deer".

Author Response

Thank you for helping us improve the quality of our work.

Reviewer’s Comment: Lines 233 and 289. "Correlations between" replace by "correlations with". 

Author’s Response: We agree with the comment. The required changes were made in the line 254 and 314. 

Reviewer’s Comment: Lines 239-240. Could you add a sentence mentioning that feed intake was not evaluated, but your approach was useful in assessing the effect of the diets? 

For example. "feed intake was not recorded, the same approach was used by some other authors (Kudrnáčová et al. 2019); therefore, our study was useful to assess the effect of β-carotene, retinol and tocopherols in the liver and the LD muscle of fallow deer". 

Author’s Response: We agree with the comment. The required sentence has been added in line 261-264.

Reviewer 2 Report (Previous Reviewer 1)

The information about the content of biologically active compounds in fallow deer meat (including β-50 carotene, retinol and tocopherols) is scarce. Therefore, the topic of the manuscript is original and interesting. Although the manuscript was revised, however, certain improvements should be made.

It is common to report the differences between the groups when those differences are statistically significant, however, even very small differences in the amounts of vitamins and provitamins can affect metabolism. I would suggest avoiding point-blank statements about in not significant differences between the groups in the Results section. More consideration should be given to the p-values ​​presented in the tables.

Lines 190-191 “The  meat samples contained around 200-210 g/kg of protein and 2-9 g/kg of fat (P=0.002).“-The results presented in Table 4 show a significant difference only for fat, therefore, (P=0.002) or (P<0.01) should be transferred to the next sentence “The concentration of EE in LD muscle samples was three- to four-fold higher (P<0.01) in the control  group (HO) than in groups HOP and HP (Table 4).“ Moreover, the first mentioned sentence is repeated in the Discussion section (Lines 296-297).

Detailed not significant differences of tocopherols between the groups presented in the Results section (Lines192-197) are also repeated in the Discussion section (Lines 310-313). The authors should choose one location of this data and put these sentences either in Results or Discussion sections. Because part of the text describing the results is repeated in the Discussion section. Therefore, I would suggest that only statistically significant and most important differences should be emphasized in the Results section. The Discussion section should comprise only those results that were not presented in the Results section.

Line 357 in Conclusions “muscle or liver“ should be “muscle and liver“

Author Response

Reviewer’s Comment: The information about the content of biologically active compounds in fallow deer meat (including β-50 carotene, retinol and tocopherols) is scarce. Therefore, the topic of the manuscript is original and interesting. Although the manuscript was revised, however, certain improvements should be made. 

Author’s Response: Thank you for helping us improve the quality of our work.

Reviewer’s Comment: It is common to report the differences between the groups when those differences are statistically significant, however, even very small differences in the amounts of vitamins and provitamins can affect metabolism. I would suggest avoiding point-blank statements about in not significant differences between the groups in the Results section. More consideration should be given to the p-values presented in the tables. 

Author’s Response: We have followed the Reviewer's comments and improved the Results section. Sentences repeating themselves in the Discussion section have been removed. Changes have also been made according to the commentary below.

Reviewer’s Comment: Lines 190-191 “The  meat samples contained around 200-210 g/kg of protein and 2-9 g/kg of fat (P=0.002).“-The results presented in Table 4 show a significant difference only for fat, therefore, (P=0.002) or (P<0.01) should be transferred to the next sentence “The concentration of EE in LD muscle samples was three- to four-fold higher (P<0.01) in the control  group (HO) than in groups HOP and HP (Table 4).“ Moreover, the first mentioned sentence is repeated in the Discussion section (Lines 296-297). 

Author’s Response: Since the sentence is repeated in the Discussion section, it has been removed from the Results section.

Reviewer’s Comment: Detailed not significant differences of tocopherols between the groups presented in the Results section (Lines192-197) are also repeated in the Discussion section (Lines 310-313). The authors should choose one location of this data and put these sentences either in Results or Discussion sections. Because part of the text describing the results is repeated in the Discussion section. Therefore, I would suggest that only statistically significant and most important differences should be emphasized in 

This manuscript is a resubmission of an earlier submission. The following is a list of the peer review reports and author responses from that submission.

Round 1

Reviewer 1 Report

As the information about the content of biologically active compounds in fallow deer meat (including β-50 carotene, retinol and tocopherols) is scarce the topic of the manuscript is original and interesting.

All the groups of deer received the same amount of hay (Table 1 in Materials and Methods section). If in the HOP group, 20% of oats DM was replaced with soybean+rapeseed meal, the entire diet did not become protein concentrates. Therefore, I would suggest to clarify the description of the groups, because reading other parts of the text (e.g. the abstract) may give the impression that the deer were fed only concentrates when practically hay in all the groups was supplemented with oat grain (HO), oat grain and soybean+rapeseed meal (HOP) and pellets. However, the composition of pellets is unclear and should be indicated. It should be described.

References to the own data and tables, which are very often repeated (Tables 3,2, 4 in lines 209, 219, 231, 236, 235, 297, 271, 281, 288, 295 and 318) in the Discussion section, are redundant. The obtained results and especially p-value (L 270-271) should be presented in the Results section.

Author Response

Thank you for your comments. We are sure that they will help us to increase the quality of our work. We will do our best to respond comprehensively and accurately. 

  1. All the groups of deer received the same amount of hay (Table 1 in Materials and Methods section). If in the HOP group, 20% of oats DM was replaced with soybean+rapeseed meal, the entire diet did not become protein concentrates. Therefore, I would suggest to clarify the description of the groups, because reading other parts of the text (e.g. the abstract) may give the impression that the deer were fed only concentrates when practically hay in all the groups was supplemented with oat grain (HO), oat grain and soybean+rapeseed meal (HOP) and pellets. However, the composition of pellets is unclear and should be indicated. It should be described.” 

Authors’ Response: We agree with your comment. We have clarified the description of the experimental groups throughout the text. Changes have been made in Simple summary (line 17), Abstract (line 26-27), Introduction (line 66), Materials and Methods (lines 81, 82, 84). 

“However, the composition of pellets is unclear and should be indicated. It should be described.” 

Authors’ Response: We agree with your comment. We have posted the raw material composition of the pellet under Table 1 in the Materials and Methods section (lines 104-106). The chemical composition of the pellet, on the other hand, can be found in Table No. 2 in the Materials and Methods section. 

  1. “References to the own data and tables, which are very often repeated (Tables 3,2, 4 in lines 209, 219, 231, 236, 235, 297, 271, 281, 288, 295 and 318) in the Discussion section, are redundant. The obtained results and especially p-value (L 270-271) should be presented in the Results section.” 

Authors’ Response: We agree with your comment. The references in the lines mentioned have been removed (lines 254, 373, 378, 520 and others have been removed in the process of changing the text as required by another Reviewer). The described results from the Discussion section have been presented in the Results section (L 191).  

Reviewer 2 Report

This study concerns the effect of the inclusion of different concentrates in the feed ration on the content of tocopherols, carotene and retinol in the liver and longissimus dorsi muscle of farm-raised fallow deer (Dama dama L.). The study is interesting, and well design and the content is relevant.

However, the works showed limits that should be addressed to be suitable for publication. In particular, the methodology should be improved. I suggest additional statistical analysis. Contrast analysis (HOP vs. HO, HP vs. HO) to evaluate the effect of the concentrates. In addition, some correlation analysis could help to justify the liver and muscle composition based on the composition of the diets. Authors should also have shown data on intakes of the different diets and (if available) nutrient intakes.

Minor comments

Methods

Lines 78-79. How was the experiment conducted? Could you provide more details?

Lines 86-87. Samples were taken for what?

Lines 109-110. How many samples were collected? From the 12 animals? How long were the samples kept, on average?

Line 192. Please revise P<0.066. The difference is not statistically significant.

Few statistical analysis was made. Correlation between diet composition and meat/liver composition should be implemented and could help to justify some of your results.

Discussion

The discussion is well written however looks like a literature review. For example, the concentration of bioactive compounds in the liver was only compared to concentrations in other animal species. The effect of the diet was not discussed. Authors should discuss the results of their study.

Lines 287-288. You should also report some correlations.

Lines 330-333. Again, you should provide some correlations

References

Some references are old, could you update them with recent literature

Author Response

Thank you for your comments. We are assured that they will help us improve the quality of our work. We will do our best to respond comprehensively and accurately. 

“In particular, the methodology should be improved. I suggest additional statistical analysis. Contrast analysis (HOP vs. HO, HP vs. HO) to evaluate the effect of the concentrates. In addition, some correlation analysis could help to justify the liver and muscle composition based on the composition of the diets.” 

Authors’ Response: We understand the Reviewer's concern about statistical issues. We know that the results and the way they are compiled and presented are an important part of the publication. When planning this experiment, as well as its methodology, we had in mind how to develop and present the obtained results in a clear, lucid and understandable way. We were looking for answers to simple research questions: what is the chemical composition of meat and liver of fallow deer fed the most popular rations used in practical feeding of fallow deer (hay and oats)? In particular, what is the content of tocopherols, retinol and is there b-carotene in the meat/liver of fallow deer fed in this way? What is the chemical composition of the meat and liver of fallow deer fed less common rations, with a slightly higher proportion of concentrated feeds (concentrate, pellets)? In particular, what will be the content of b-carotene, tocopherols, retinol in the meat/liver of fallow deer fed this way? The aforementioned questions seemed to us interesting and reasonable from a practical point of view, but also from a scientific point of view, since such information is not found in the literature. 

In view of the above, we agree that checking the correlation between the studied compounds will be an additional, positive point to answer the research questions, so we have performed such analyses (Table 5, Table 6, L221-234). However, the inclusion of contrast analysis in this manuscript, in the opinion of the authors, is unjustified and not necessarily to answer the research questions posed, so no such analysis has been included. 

“Authors should also have shown data on intakes of the different diets and (if available) nutrient intakes.” 

Authors’ Response: During the planning of the experiment, the assumption was that the animals were kept in groups (each experimental group was kept in a separate box, with 12 animals per box), which eliminates the possibility of controlling feed intake separately for each animal. It was only possible to control feed intake collectively for each experimental group, and then possibly estimate intake for each individual. However, such data would be subject to error. The mentioned data were not collected during the experiment, as the Authors did not consider it necessary to answer the research questions.   

Minor comments 

Methods 

  1. “Lines 78-79. How was the experiment conducted? Could you provide more details?” 

Authors’Response: Does the Reviewer have a more detailed description of the adaptation period in mind? If so, the description is as follows: the entire experience lasted 60 days. The first 21 days of the experiment was the adaptation period. This is the time when the animals are fed the experimental rations (hay+oats; hay+oats+protein concentrate; hay+pelet), in order to adapt the intestinal flora to the experimental ration and to clean the digestive tract from the feeds used before the experiment, as well as to fill the digestive tract with the experimental feeds. The animal body during this period becomes accustomed to digesting the experimental feeds. After the initial period, feeding with experimental rations continues until the experiment is completed and the animals from which we took meat samples are slaughtered.   

If more information is needed - please let me know and I will attach it immediately. 

  1. “Lines 86-87. Samples were taken for what?” 

Authors’ Response: Representative samples of rations were taken before feeding the animals to determine their chemical composition (L 88), which is listed in Table 2 in the Materials and Methods section.  

  1. “Lines 109-110. How many samples were collected? From the 12 animals? How long were the samples kept, on average?” 

Authors’ Response: Meat and liver samples were taken from each animal in each experimental group (3x12=36 samples; L 114). The meat and liver samples were frozen for about one weeks before analysis (L 116). 

  1. “Line 192. Please revise P<0.066. The difference is not statistically significant.” 

Authors’ Response: We agree with your comment. P<0.066 has been removed (L 200). 

  1. “Few statistical analysis was made. Correlation between diet composition and meat/liver composition should be implemented and could help to justify some of your results.” 

Authors’ Response: We conducted a correlation analysis (table 5, table 6). 

Discussion 

  1. The discussion is well written however looks like a literature review. For example, the concentration of bioactive compounds in the liver was only compared to concentrations in other animal species. The effect of the diet was not discussed. Authors should discuss the results of their study. 

Authors’ Response: Changes in the Discussion have been made (L 242-251; 264-346; 352-353; 356-356; 371-372; 384-386; 501-512; 520; 531). 

  1. “Lines 287-288. You should also report some correlations.” 

Authors’ Response: Correlations have been added (L 384-386). 

  1. Lines 330-333. Again, you should provide some correlations.” 

Authors’ Response: Correlations have been added (L 531). 

References 

  1. Some references are old, could you update them with recent literature” 

Authors’ Response: We agree with your comment, but not completely. We agree that it is reasonable to use the latest literature when writing a manuscript. We are aware that many factors involved in conducting scientific research are dynamically evolving. However, authors know the risks of using publications from previous decades and knowingly take them. In some cases, this is dictated by necessity, since knowledge of the present topic has not always been updated in recent years. In addition, our observation is that scientific studies involving carotenoids, tocopherols and retinols are characterized by a periodic, wave-like occurrence - first around the '20s, then in the '50s, then in the '80s and now. Secondly, the authors have often purposely referred to publications from previous decades, as we believe that their merit is often greater than those published in previous years. Accordingly, we have replaced some publications issued in previous decades with newer ones (number 2, 13, 26, 42, 51). However, we left some of them. 

Reviewer 3 Report

The experiment is scientifically sound and the results and discussion were presented with good clarity. However, the conclusion is too wordy (L324 - 335: The conclusion is too long and should be precisely summarised).

Author Response

Thank you for your comment and for taking the time to read our work. Your review will help us to increase the quality of our work.   

“However, the conclusion is too wordy (L324 - 335: The conclusion is too long and should be precisely summarised).” 

Authors’ Response: We agree with your comment, the Conclusion section can indeed be written more concisely without losing substance. We have followed your comment and shortened this paragraph (L526-532). 

Round 2

Reviewer 2 Report

The authors reply to most of our queries. However, there is still a major issue.

The lack of intake (DM and Nutrient) data is a serious flaw in this study. Feed intakes should have been recorded, allowing for assessment of the real effect of the experimental diets.

Please see attached other comments.
